# Adopting telehealth for mental health support in stroke rehabilitation in Sub-Saharan Africa: A conceptual analysis using Rogers' Diffusion of Innovation Theory

Delight Tsogbe©*, Tettey Martin©, Akuffo Bernice Gyanmea©

Department of Population and Behavioral Sciences, Fred N. Binka School of Public Health, University of Health and Allied Sciences, Ho, Ghana

* delighttsogbe@gmail.com

## Abstract

Stroke rehabilitation in Sub-Saharan Africa (SSA) faces persistent gaps in mental health service integration. Post-stroke depression, anxiety, and cognitive impairments remain underdiagnosed due to workforce shortages, stigma, and infrastructural constraints. Telehealth technologies offer a potential mechanism to expand access to psychological care, yet adoption across SSA remains uneven. This paper provides a conceptual, theory-driven synthesis exploring telehealth adoption for mental health support in stroke rehabilitation across SSA, guided by Rogers' Diffusion of Innovation Theory. A structured conceptual review was conducted using literature from PubMed, Scopus, AJOL, and Google Scholar (2005–2024). Inclusion criteria focused on SSA studies addressing telehealth or mHealth in mental health or rehabilitation contexts. Thematic synthesis was guided by Rogers' five innovation attributes, relative advantage, compatibility, complexity, trialability, and observability, to assess adoption facilitators and barriers. Telehealth adoption is influenced by cultural compatibility, simplicity of design, perceived relative advantage, observable benefits through pilot projects, and supportive policy environments. Successful initiatives in Ghana, Kenya, Nigeria, and Uganda demonstrate the importance of localized adaptation and user engagement. Telehealth provides a scalable pathway to address post-stroke mental health needs in SSA. Effective adoption requires culturally responsive design, targeted capacity-building, and policy alignment with national and regional digital health strategies.

## Introduction

Stroke remains one of the leading causes of disability and mortality in Sub-Saharan Africa (SSA). The mental health sequelae of stroke, such as depression, anxiety, and cognitive impairment, significantly reduce recovery outcomes. However, mental

---

---

**Data availability statement:** This study is a conceptual and theory-driven synthesis based on previously published literature. All data supporting the findings of this study are derived from publicly available sources cited within the manuscript's reference list. No new datasets were generated or analyzed during the current study.

**Funding:** The authors received no specific funding for this work.

**Competing interests:** The authors have declared that no competing interests exist.

health services are severely limited due to infrastructural barriers, insufficient specialists, and sociocultural stigma [1,2].

Telehealth, encompassing video consultations, mobile health (mHealth), and remote rehabilitation, has emerged as a potential solution to overcome geographic and workforce barriers [3]. Despite increasing attention, telehealth adoption in SSA remains inconsistent, constrained by cost, digital literacy, and fragmented policy environments [4].

To understand these challenges conceptually, this paper applies Rogers' Diffusion of Innovation Theory [5] to synthesize evidence on telehealth adoption for mental health support in stroke rehabilitation across SSA.

## Methods

### Conceptual design

This study adopts a theory-driven conceptual synthesis approach using Rogers' Diffusion of Innovation Theory as the analytical framework. The approach integrates existing empirical and theoretical evidence to derive implementation insights rather than generating primary data.

### Theoretical framework

Rogers' Diffusion of Innovation Theory provides a systematic lens for understanding how new technologies, such as telehealth, are adopted within health systems. The theory categorizes adopters into five groups, innovators, early adopters, early majority, late majority, and laggards, each differing in their readiness to embrace innovation. It also identifies five key innovation attributes that influence adoption decisions: relative advantage (the perceived improvement over existing methods), compatibility (alignment with users' values and needs), complexity (ease of use), trialability (the ability to experiment on a limited basis), and observability (visibility of benefits and outcomes). Applying this framework offers a structured understanding of how telehealth can be effectively adopted to provide mental health support during stroke rehabilitation across Sub-Saharan Africa, highlighting the social, technological, and contextual factors that shape diffusion dynamics. A schematic representation of these five innovation attributes and their relationship to telehealth adoption in stroke rehabilitation is illustrated in Fig 1.

### Search strategy

Databases searched included PubMed, Scopus, AJOL, and Google Scholar between 2005 and 2024. Search terms included:

"telehealth", "mental health", "stroke rehabilitation", "Sub-Saharan Africa", "digital health", and "Diffusion of Innovation Theory".

### Inclusion criteria

• Focus on Sub-Saharan Africa

• Telehealth or mHealth interventions for mental health or stroke rehabilitation

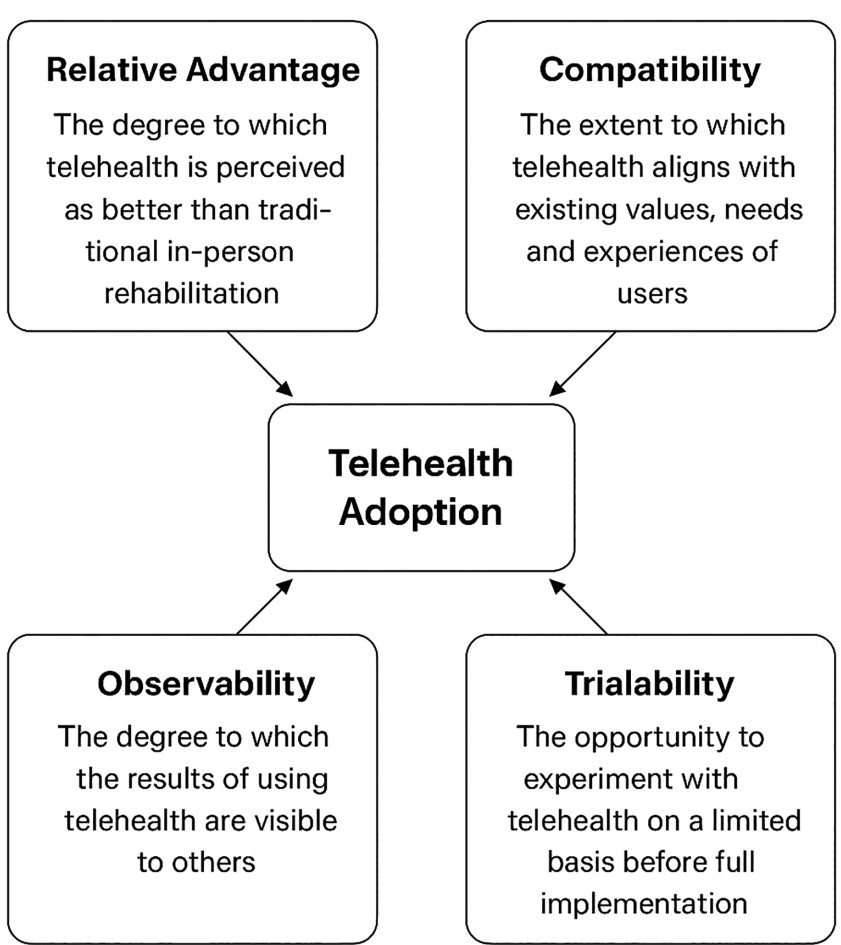

**Fig 1. The five innovation attributes influencing telehealth adoption in stroke rehabilitation (adapted from Rogers, 2003).**

- Articles discussing implementation, barriers, or adoption outcomes

- Peer-reviewed or credible institutional reports

**Exclusion criteria**

- Non-SSA contexts

- Studies unrelated to telehealth or mental health

- Commentary pieces lacking implementation evidence

**Data extraction and synthesis**

Eligible studies were screened manually and coded under Rogers' five innovation attributes. Key data extracted included country, intervention type, health condition, findings, and coded attribute. Reporting transparency followed ENTREQ and PRISMA-ScR recommendations. A summary of the reviewed studies across Sub-Saharan Africa is presented in Table 1.

**Table 1. Summary of Reviewed Studies (2005–2024).**

| Country | Intervention Type | Focus Area | Key Findings | Innovation Attribute |
|---|---|---|---|---|
| Ghana | Telepsychiatry pilot (mHealth) | Post-stroke depression | Improved follow-up and access | Compatibility |
| Kenya | Mobile rehab platform | Stroke rehabilitation continuity | Feasible, low dropout rate | Trialability |
| Nigeria | Video-based teletherapy | Mental health counselling | Effective but limited by bandwidth | Complexity |
| South Africa | Hybrid telehealth for chronic care | Cognitive rehab post-stroke | Improved adherence | Relative Advantage |
| Uganda | SMS mental health support | Depression screening | Enhanced patient engagement | Observability |

## Results and discussion

### Compatibility

Telehealth initiatives aligning with local clinical routines and cultural contexts show better uptake. Culturally adapted interventions, such as Ghana's community telepsychiatry model, demonstrated higher engagement due to language inclusivity and integration with existing health infrastructure [6]. Lack of alignment with cultural norms or poor stakeholder involvement often hinder scalability [7].

### Complexity

Adoption depends heavily on technology usability and digital literacy. Simplified platforms with training modules have succeeded in Kenya and Ethiopia [8]. Conversely, high system complexity, unstable connectivity, and English-only interfaces create barriers for rural populations [9].

### Relative advantage

Perceived benefits, such as reduced travel costs, improved access to specialists, and enhanced continuity of care, drive adoption. Evidence from South Africa and Nigeria indicates that telehealth can complement face-to-face models by reducing logistical burdens [10,11].

### Trialability and observability

Pilot programs in Ghana and Uganda improved stakeholder trust by allowing experimentation in controlled settings. Observable benefits, such as improved therapy attendance and patient satisfaction, further encouraged diffusion [12]. Documented visibility of success is a strong determinant of scaling efforts across SSA [13].

### Critical appraisal of evidence

Most studies reviewed were small-scale or pilot interventions, limiting generalizability. Few reported long-term outcomes or cost-effectiveness analyses. The evidence base remains fragmented, highlighting the need for empirical studies that measure clinical outcomes, sustainability, and economic viability.

### Equity and ethical considerations

Digital health adoption in SSA must address digital exclusion, particularly among rural women, the elderly, and low-literacy groups [14]. Ethical concerns around data privacy, informed consent, and trust are also significant, given limited digital governance structures. Culturally sensitive design and strong data protection frameworks are essential for equitable telehealth expansion.

## Policy and implementation recommendations

• Integrate telehealth within national eHealth and UHC frameworks.

• Establish digital literacy training for health workers and patients.

• Support pilot-to-scale pathways through public-private partnerships.

• Ensure affordable digital infrastructure and regulatory oversight.

• Align strategies with African Union's Digital Health Strategy (2023).

## Conclusion

Telehealth represents a transformative opportunity to strengthen post-stroke mental health rehabilitation in SSA. Adoption success depends on alignment with cultural and infrastructural realities, simplicity of design, visible benefits, and supportive policy ecosystems. A systems-level, equity-focused approach can ensure that telehealth advances inclusion rather than exacerbating digital divides.

## Limitations

This conceptual synthesis draws on secondary literature and does not include primary data collection. Findings depend on the quality and availability of existing evidence, which varies across SSA contexts.

## Author contributions

**Data curation:** Delight Tsogbe.

**Methodology:** Delight Tsogbe.

**Project administration:** Tettey Martin.

**Resources:** Tettey Martin.

**Visualization:** Akuffo Bernice Gyanmea.

**Writing – original draft:** Delight Tsogbe.

**Writing – review & editing:** Delight Tsogbe, Tettey Martin, Akuffo Bernice Gyanmea.

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
