## [Decision Letter · Decision Letter 0]

9 Oct 2025

PGPH-D-25-02080

Adopting Telehealth for Mental Health Support in Stroke Rehabilitation in Sub-Saharan Africa: A Conceptual Analysis Using Rogers’ Diffusion of Innovation Theory

Dear Dr. TSOGBE,

Thank you for submitting your manuscript to PLOS Global Public Health. After careful consideration, we feel that it has merit but does not fully meet PLOS Global Public Health’s publication criteria as it currently stands. Therefore, we invite you to submit a revised version of the manuscript that addresses the points raised during the review process.

We look forward to receiving your revised manuscript.

Kind regards,

Prof Razak Gyasi, PhD, PD

Academic Editor

Journal Requirements:

1. Please send a completed 'Competing Interests' statement, including any COIs declared by your co-authors. If you have no competing interests to declare, please state "The authors have declared that no competing interests exist". Otherwise please declare all competing interests beginning with the statement "I have read the journal's policy and the authors of this manuscript have the following competing interests:"

3. We ask that a manuscript source file is provided at Revision. Please upload your manuscript file as a .doc, .docx, .rtf or .tex.

4. Please provide separate figure files in .tif or .eps format.

Reviewers' comments:

Reviewer's Responses to Questions

**Comments to the Author**

1. Does this manuscript meet PLOS Global Public Health’s publication criteria?

Reviewer #1: Yes

Reviewer #2: Yes

Reviewer #3: No

Reviewer #4: Partly

2. Has the statistical analysis been performed appropriately and rigorously?

Reviewer #1: N/A

Reviewer #2: N/A

Reviewer #3: No

Reviewer #4: No

3. Have the authors made all data underlying the findings in their manuscript fully available (please refer to the Data Availability Statement at the start of the manuscript PDF file)?

Reviewer #1: Yes

Reviewer #2: Yes

Reviewer #3: No

Reviewer #4: Yes

4. Is the manuscript presented in an intelligible fashion and written in standard English?

Reviewer #1: Yes

Reviewer #2: Yes

Reviewer #3: No

Reviewer #4: Yes

Reviewer #1: Review of Manuscript: Telehealth for Mental Health Support During Stroke Rehabilitation in Sub-Saharan Africa: A Conceptual Analysis Using Rogers’ Diffusion of Innovation Theory

General Assessment

This manuscript presents a theory-driven conceptual synthesis examining how telehealth interventions could support mental health during stroke rehabilitation in Sub-Saharan Africa (SSA). By applying Rogers’ Diffusion of Innovation Theory, the authors analyze opportunities and barriers to telehealth adoption. The paper is well-motivated, addresses a significant gap in public health, and is generally clear and accessible. However, as this is not an empirical study but a conceptual review, the scope, methodology, and claims need to be carefully calibrated to align with PLOS Global Public Health standards.

Evaluation Against Editorial Criteria

1. Original Research

The article presents conceptual and synthesis-based research rather than empirical primary data collection. While conceptual analyses can be original and valuable, the framing should be explicitly described as a narrative/theoretical synthesis. Currently, some sections imply empirical rigor (e.g., “methods” structured like a systematic review), which may confuse readers about the nature of the evidence.

Recommendation: Clarify the scope of originality and emphasize that this as a conceptual/theory-driven contribution that synthesizes available evidence.

2. Novelty / Prior Publication

The results do not appear to be published elsewhere. No concerns here.

3. Technical Rigor & Methodological Transparency

The methods are described as a “structured narrative review” guided by Rogers’ framework. Inclusion and exclusion criteria are mentioned, but search and selection procedures lack reproducibility (e.g., no PRISMA flowchart, no list of final included studies). The categorization of findings under the five attributes is reasonable but could be more systematic.

Recommendations:

• Provide a table summarizing included studies (country, population, intervention, findings, attribute coded).

• Specify how many articles were retrieved, screened, and finally included.

• Consider referencing relevant reporting standards (e.g., ENTREQ for qualitative evidence synthesis).

4. Validity of Conclusions

The conclusions, that telehealth is promising but requires cultural adaptation, training, and policy integration, are well-supported by the thematic synthesis. However, the strength of evidence is variable since it relies heavily on small pilot programs and conceptual reasoning. The authors should temper claims by highlighting the paucity of robust empirical evidence in SSA.

Recommendation: Add explicit caveats that the findings are primarily conceptual and based on limited empirical data.

5. Clarity of Writing

The manuscript is written in clear and intelligible English. Minor grammatical inconsistencies exist (e.g., “five type’s” instead of “five types”). The flow between theory, methods, findings, and discussion is logical, though the paper occasionally repeats theoretical definitions.

Recommendation: Minor copyediting to improve grammar and eliminate redundancy.

6. Ethical Standards

No ethical concerns arise, as this is not human-subject research. Declarations of ethics approval, consent, and competing interests are appropriately noted.

7. Reporting Standards & Data Availability

• The article does not include primary data but synthesizes published studies.

• Reporting standards such as PRISMA for scoping reviews or ENTREQ could be referenced to strengthen transparency.

• No data repository is applicable here, but a supplementary appendix listing included studies would enhance rigor.

Contextualization and Literature

The authors situate their arguments within relevant telehealth and adoption literature, citing both global and SSA-specific sources. However, some references are outdated or not region-specific (e.g., reliance on general telemedicine reviews). There is room to engage more deeply with SSA-based digital health scholarship.

Recommendation: Expand references to include recent SSA-focused telehealth evaluations (post-2019), particularly in Ghana, Nigeria, Kenya, and South Africa.

Significance for the Discipline

The paper highlights an underexplored but highly relevant topic: integrating digital tools into stroke rehabilitation and mental health care in SSA. It contributes by applying a structured theoretical framework, which could guide future empirical research and policy interventions. However, without stronger methodological transparency, its impact may be limited.

Recommendation

Decision: Major Revision

The manuscript addresses an important gap and has the potential to be published in PLOS Global Public Health after revisions. The main issues are:

1. Clarify that this is a conceptual/theory-driven synthesis, not empirical research.

2. Strengthen methodological transparency with structured reporting of search and study inclusion.

3. Provide summary tables/appendices of reviewed studies.

4. Temper claims to reflect limited empirical evidence.

5. Minor copyediting and grammar corrections.

6. Expand engagement with recent SSA telehealth literature.

With these revisions, the paper could make a meaningful contribution to the field.

Reviewer #2: The manuscript is presented in the way to make easily understanding of the findings and the conclusions are stated appropriate. In case this abstract has been accepted for publication I encourage the authors to its wide diffusion. I did not register any potential competing interests on the part of the authors, data availability, or research ethics.

Reviewer #3: Topic of the manuscript is unique which is good but it shall be difficult for the reader to conceptualize how the author came up with this conclusion in the absence of proper methodology. While reading the paper, there is a confusion between conceptual commentary or a narrative or systematic analysis. The paper says, conceptual analysis but it lacks transparent search strategy, appropriate analysis, and clear exclusion and inclusion criteria.

Reviewer #4: The topic is highly relevant, addressing a pressing health system challenge in low-resource contexts. The manuscript is clearly written and accessible to a multidisciplinary readership.

However, the current version is limited by its largely conceptual orientation, limited methodological transparency, and insufficient critical engagement with existing empirical evidence. The paper would benefit from a stronger methodological account, deeper exploration of practical and equity implications, and clearer policy recommendations for stakeholders in SSA.

Review Comments-

1. Methodological Transparency

The methods section currently provides limited information on how the literature was identified and synthesised. For reproducibility, readers need details of databases searched, timeframe, inclusion/exclusion criteria, and number of articles reviewed.

Even if framed as a conceptual synthesis, adopting elements of PRISMA reporting would increase rigour.

2. Critical Engagement with Evidence

The findings are presented descriptively (e.g., “pilot studies showed promising outcomes”) but lack critical appraisal of study quality, effect sizes, or sustainability.

Please incorporate discussion of strengths and weaknesses of the underlying studies, including why some interventions succeeded or failed.

3. Equity and Ethical Dimensions

While digital adoption challenges are mentioned, the manuscript underplays issues such as digital exclusion (e.g., rural women, older adults, low-literacy groups) and ethical concerns around privacy, data ownership, and trust.

Expanding on these issues would strengthen alignment with PLOS Global Public Health’s equity-focused mission.

4. Policy and Implementation Implications

The conclusion is currently broad (e.g., “strategies should be developed”), without concrete pathways.

Please provide specific, actionable recommendations for policymakers, ministries of health, and digital health funders in SSA — for example, integration into UHC frameworks, African Union digital health strategies, or workforce training models.

5. Balance Between Theory and Practice

While Rogers’ Diffusion of Innovation Theory structures the analysis well, the paper risks being overly abstract.

Consider adding case vignettes or short examples from actual SSA telehealth implementations (e.g., Nigeria, Kenya, South Africa during COVID-19). This will ground the conceptual analysis in lived practice.

6. References: Several citations are dated or not SSA-specific (for e.g., Davis 1989; Czaja 2006). Please incorporate more recent African digital health literature, including WHO Africa telehealth initiatives and COVID-era studies. Try to include more references to provide a wide range of literature to the reader.

7. Limitations: Explicitly acknowledge that findings are derived from secondary sources, not primary empirical data.

**Do you want your identity to be public for this peer review?** For information about this choice, including consent withdrawal, please see our Privacy Policy

Reviewer #1: **Yes:** Ivan Calder

Reviewer #2: **Yes:** Placido Monteiro Cardoso,MD,Msd,D.sc, Guinea-Bissau

Reviewer #3: No

Reviewer #4: **Yes:** Dr. Ratendra Chauhan

---

## [Decision Letter · Decision Letter 1]

5 Jan 2026

Adopting Telehealth for Mental Health Support in Stroke Rehabilitation in Sub-Saharan Africa: A Conceptual Analysis Using Rogers’ Diffusion of Innovation Theory

PGPH-D-25-02080R1

Dear Mr TSOGBE,

We are pleased to inform you that your manuscript 'Adopting Telehealth for Mental Health Support in Stroke Rehabilitation in Sub-Saharan Africa: A Conceptual Analysis Using Rogers’ Diffusion of Innovation Theory' has been provisionally accepted for publication in PLOS Global Public Health.

Best regards,

Professor Razak Gyasi, PhD, PD

Academic Editor

Reviewer Comments (if any, and for reference):

Reviewer's Responses to Questions

**Comments to the Author**

Reviewer #1: All comments have been addressed

Reviewer #2: All comments have been addressed

publication criteria?

Reviewer #1: Yes

Reviewer #2: Yes

3. Has the statistical analysis been performed appropriately and rigorously?

Reviewer #1: N/A

Reviewer #2: N/A

4. Have the authors made all data underlying the findings in their manuscript fully available (please refer to the Data Availability Statement at the start of the manuscript PDF file)?

Reviewer #1: Yes

Reviewer #2: No

5. Is the manuscript presented in an intelligible fashion and written in standard English?

Reviewer #1: Yes

Reviewer #2: Yes

Reviewer #1: The revised manuscript is substantially improved. The authors have clearly distinguished the work as a conceptual, theory-driven synthesis and have expanded methodological transparency (databases searched, timeframe, inclusion/exclusion criteria, coding approach). The addition of a summary table and the new sections on evidence limitations, equity/ethics, and policy recommendations strengthen the contribution and align the paper with PLOS Global Public Health priorities.

Reviewer #2: Comment 1: The rationale for selecting the 2005–2024 timeframe should be briefly clarified.

Comment 2: Equity issues are discussed but insufficiently translated into the policy recommendations.

**Do you want your identity to be public for this peer review?** For information about this choice, including consent withdrawal, please see our Privacy Policy

Reviewer #1: No

Reviewer #2: No
